



# Probabilistic modelling of substorm occurrences with an echo state network

Shin'ya Nakano[1,2,3], Ryuho Kataoka[4,3], Masahito Nosé[5], and Jesper W. Gjerloev[6]

[1]The Institute of Statistical Mathematics, Tachikawa, 190–8562, Japan
[2]Center for Data Assimilation Research and Applications, Joint Support Center for Data Science Research, Tachikawa, Japan.
[3]The Graduate University for Advanced Studies, SOKENDAI, Hayama, Japan.
[4]National Institute of Polar Research, Tachikawa, Japan.
[5]Institute for Space-Earth Environmental Research, Nagoya University, Nagoya, Japan.
[6]Johns Hopkins University Applied Physics Laboratory, Laurel, MD, USA.

**Correspondence:** Shin'ya Nakano (shiny@ism.ac.jp)

**Abstract.** The relationship between solar wind conditions and substorm activity is modelled with an approach based on an echo state network. Substorms are a fundamental physical phenomenon in the magnetosphere–ionosphere system, but the deterministic prediction of substorm onset is very difficult because the physical processes that underlie substorm occurrences are complex. To model the relationship between substorm activity and solar wind conditions, we treat substorm onset as a stochastic phenomenon and represent the stochastic occurrences of substorms with a nonstationary Poisson process. The occurrence rate of substorms is then described with an echo state network model. We apply this approach to two kinds of substorm onset proxies. One is a sequence of substorm onsets identified from auroral electrojet intensity and the other is onset events identified from Pi2 activity. We then analyse the response of substorm activity to solar wind conditions by feeding synthetic solar wind data into the echo state network. The results indicate that the effect of the solar wind speed is important, especially for Pi2 substorms. A Pi2 pulsation can often occur even if the interplanetary magnetic field (IMF) is northward, while the activity of auroral electrojets is depressed during northward IMF conditions. We also observe spiky enhancements in the occurrence rate of substorms when the solar wind density abruptly increases, which might suggest an external triggering due to a sudden impulse of solar wind dynamic pressure. It seems that northward turning of the IMF also contributes to substorm occurrences, though the effect is likely to be minor.

## 1 Introduction

Substorms are a fundamental physical phenomenon in the magnetosphere–ionosphere system. Since substorms are the main source of geomagnetic disturbances in the polar ionosphere causing geomagnetically induced currents (e.g., Viljanen et al., 2006; Wei et al., 2021; Schillings et al., 2022), prediction of substorms is an important issue. Substorms are most likely driven by the solar wind, so it is essential to understand the relationship between substorms and solar wind conditions for predicting substorms. In this context, many studies have attempted to construct predictive models of the $AE$ indices, which represent the intensities of auroral electrojets in the polar ionosphere (Davis and Sugiura, 1966). For example, Luo et al. (2013) de-



veloped a parametric model for predicting the $AE$ indices from a time history of solar-wind data. There have also been a number of studies which employed machine learning approaches for predicting the $AE$ indices from given solar-wind condi-tions (e.g., Gleisner and Lundstedy, 1997; Takalo and Timonen, 1997; Amariutei and Ganushkina, 2012). Our previous study

(Nakano and Kataoka, 2022) (hereinafter referred to as NK22) also modelled the relationship between the $AE$ indices and solar-wind variables using an echo state network (ESN) model, which is a kind of recurrent neural network originally intro-duced by Jaeger and Haas (2004).

However, a good prediction of the $AE$ indices does not necessarily guarantee successful prediction of substorm onsets. Many studies have argued that substorm onsets are triggered by internal processes of the magnetosphere–ionosphere system (e.g.,

Baker et al., 1996; Lui, 1996; Lyons et al., 2018; Miyashita and Ieda, 2018). Considering that substorms sometimes take place without any visible solar-wind variations (e.g., Nakano and Iyemori, 2005), substorms may be caused by complex magneto-spheric processes which are hard to predict. It would thus be difficult to deterministically predict individual substorm onsets from a given time history of solar-wind conditions. As a matter of fact, the ESN in our previous study NK22 did not always predict sharp decreases of the $AL$ index due to a substorm expansion. Figure 1 compares the observed Kyoto $AU$ and $AL$

indices (World Data Center for Geomagnetism, Kyoto et al., 2015) and the prediction with the ESN. Although the prediction shown by the red line roughly traces the actual $AU$ and $AL$ indices, it did not reproduce many negative $AL$ spikes.

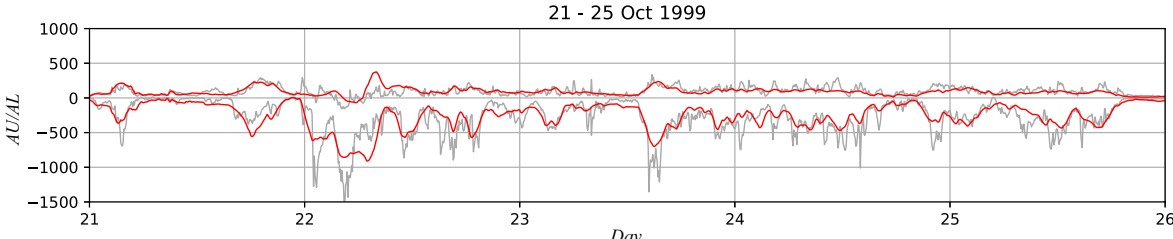

**Figure 1.** Prediction of $AU$ and $AL$ indices with the ESN in NK22 (red) and actual $AU$ and $AL$ indices calculated by the World Data Center, Kyoto (gray).

In this study, we propose another approach for modelling the relationship between substorm occurrences and solar-wind input. To deal with the complexity underlying substorm occurrences, we treat a substorm onset as a stochastic phenomenon. The stochastic occurrence of substorms is represented with a nonstationary Poisson process, which is a probabilistic model

describing event time series. The occurrence rate of substorms is then modelled with an echo state network (ESN) model. By introducing the ESN, we can represent the dependence of the occurrence rate on solar wind conditions. A study has performed short-term prediction of substorm onsets for one hour intervals with a convolutional neural network (Maimaiti et al., 2019). In contrast, our approach sequentially processes the time series of substorm events to allow analysis of the response of substorm activity to the solar wind input.

We applied the proposed ESN-based approach to two types of substorm onsets. One is a sequence of substorm onsets identified from the *SML* index (Newell and Gjerloev, 2011a, b). The *SML* index is a proxy of westward auroral electrojet in-





tensity derived from the SuperMAG geomagnetic data (Gjerloev, 2012) with the same algorithm as for the $AL$ index. However, since an increase of the auroral electrojet does not necessarily indicate a substorm onset (e.g., Kamide and Kokubun, 1996), the events determined from the auroral electrojet intensity may contain non-substorm events such as DP2 type convection enhance-

ments. To identify substorm onsets without relying on the auroral electrojet intensity, we also analyse a sequence of substorm onsets identified from the $Wp$ index (Nosé et al., 2009, 2012; World Data Center for Geomagnetism, Kyoto and Nosé, 2016). The $Wp$ index corresponds to the amplitude of low-latitude Pi2 pulsations. An onset determined from the $Wp$ index is thus regarded as a Pi2 event. After training the ESN to model the two types of substorms, the relationship between substorm activity and solar wind conditions is discussed by analysing the response of the ESN outputs to synthetic solar wind inputs.

## 2  Method

We denote $\nu(t|\boldsymbol{\beta})dt$ as the probability of the occurrence of an event within a small time interval $dt$, where $\boldsymbol{\beta}$ is a parameter determining the shape of the function $\nu$. The function $\nu$ is referred to as the intensity function and it corresponds to the instantaneous occurrence rate per unit time. Given a sequence of event occurrence times $\tau_{1:N} = \{\tau_1, \tau_2, \ldots, \tau_N\}$, the likelihood of the parameter $\boldsymbol{\beta}$ is written as follows:

$$
L(\boldsymbol{\beta}) = p(\tau_{1:N}|\boldsymbol{\beta}) = \prod_{i=1}^{N} \nu(\tau_i|\boldsymbol{\beta}) \exp\left[ -\int_{t_0}^{t_K} \nu(t|\boldsymbol{\beta})\,dt \right]
\tag{1}
$$

(e.g., Daley and Vere-Jones, 2003), where $t$ denotes time and $N$ is the number of events. The log-likelihood thus becomes

$$
\log L(\boldsymbol{\beta}) = \sum_{i=1}^{N} \log \nu(\tau_i|\boldsymbol{\beta}) - \int_{t_0}^{t_K} \nu(t|\boldsymbol{\beta})\,dt.
\tag{2}
$$

Discretising Eq. (2) in time, we obtain

$$
\log L(\boldsymbol{\beta}) = \sum_{i=1}^{N} \log \nu(\tau_i|\boldsymbol{\beta}) - \sum_{k=0}^{K-1} \nu(t_k|\boldsymbol{\beta})\Delta t,
\tag{3}
$$

where $t_k = t_0 + k\Delta t$. The time interval $\Delta t$ in Eq. (3) is set to be 5 minutes in this study. If an event occurs at $t = \tau_i$ during large $\nu$, the first term on the righthand side of Eq. (2) gets larger. On the other hand, the second term on the righthand side of Eq. (2) falls down if $\nu$ keeps large values for whole interval. The log-likelihood $\log L$ thus becomes larger by choosing an intensity function which becomes large when events frequently occur and small when events do not occur. We want to obtain a good intensity function which describes the occurrence rate of events better.

We model the function $\nu$ with the ESN (Jaeger and Haas, 2004), used in NK22 and another recent study (Kataoka and Nakano, 2021). Denoting the vector consisting of the state variables of the ESN at time $t_k$ as $\boldsymbol{x}_k$, the $i$-th element of $\boldsymbol{x}_k$, $x_{k,i}$ is updated at each time step according to the following equation:

$$
x_{k,i} = \tanh\left( \boldsymbol{w}_i^\mathsf{T} \boldsymbol{x}_{k-1} + \boldsymbol{u}_i^\mathsf{T} \boldsymbol{z}_k + \eta_i \right), \quad (i = 1, \ldots, m)
\tag{4}
$$





where $\boldsymbol{z}_k$ is the vector of the input variables, $\boldsymbol{w}_i$ is the vector of the weights connecting the state variables, $\boldsymbol{u}_i$ is the vector of

the weights connecting the input variables and state variables, and $m$ denotes the dimension of the vector $\boldsymbol{x}_k$. The dimension $m$ is set to be 1000 in this study. The input vector $\boldsymbol{z}_k$ is given as follows:

$$
\boldsymbol{z}_k =
\begin{pmatrix}
B_{x,k}/S_{B_x} \\
B_{y,k}/S_{B_y} \\
B_{z,k}/S_{B_z} \\
(V_{sw,k} - b_V)/S_V \\
(N_{sw,k} - b_N)/S_N \\
(\Theta_{sw,k} - b_\Theta)/S_\Theta \\
\cos(2\pi H_k/24) \\
\sin(2\pi H_k/24) \\
\cos(2\pi D_k/364.24) \\
\sin(2\pi D_k/364.24)
\end{pmatrix}
\tag{5}
$$

where $B_{z,k}$, $B_{y,k}$, $B_{x,k}$ respectively denote the $z$, $y$, and $x$ components of the interplanetary magnetic field in the geocentric solar magnetic (GSM) coordinates at time $t_k$, $V_{sw,k}$ is the $-x$ component of the solar wind velocity in the GSM coordinates,

and $N_{sw,k}$ and $\Theta_{sw,k}$ are the solar wind density and temperature, respectively. These solar wind variables are taken from the OMNI 5-minute data. $H_k$ and $D_k$ respectively indicate universal time (UT) in hours and the day from the end of 2000 (i.e., $D_k = 1$ on January 1, 2001) for determining the UT dependence and seasonal dependence (e.g., Cliver et al., 2000). $S_{B_x}$, $S_{B_y}$, $S_{B_z}$, $S_V$, $S_N$, and $S_T$ are rescaling factors to adjust the value of each element of $\boldsymbol{z}_k$ to a similar range, and $b_V$, $b_N$, and $b_T$ are for adjusting the range of each element of $\boldsymbol{z}_k$. According to NK22, we assume $S_{B_x} = S_{B_y} = S_{B_z} = 10\,(\mathrm{nT})$,

$S_V = 500\,(\mathrm{km/s})$, $S_N = 20\,(/\mathrm{cc})$, $S_T = 10^6\,(\mathrm{K})$, $b_V = 400\,(\mathrm{km/s})$, $b_N = 1\,(/\mathrm{cc})$, and $b_T = 2 \times 10^5\,(\mathrm{K})$. The weights $\{\boldsymbol{w}_i\}$ and $\{\boldsymbol{u}_i\}$ are randomly given in advance and are fixed. The parameters $\{\eta_i\}$ are also randomly given and are fixed. Specifically, we randomly chose 90% of the weights $\{\boldsymbol{w}_i\}$ and $\{\boldsymbol{u}_i\}$ and set them to 0. The non-zero elements of $\boldsymbol{u}_i$ and $\eta_i$ were given randomly from a standard normal distribution. The non-zero elements of $\boldsymbol{w}_i$ were also drawn from a normal distribution. The weights $\{\boldsymbol{w}_i\}$ were then rescaled such that the maximum singular value of the weight matrix is 0.99, where the weight matrix

$\mathbf{W}$ is an $m \times m$ matrix defined as

$$
\mathbf{W} = (\boldsymbol{w}_1 \; \boldsymbol{w}_2 \; \cdots \boldsymbol{w}_m).
\tag{6}
$$

By setting the maximum singular value of $\mathbf{W}$ to be less than unity, it is guaranteed that the ESN forgets distant past inputs and that the weights can be stably determined. Although NK22 employed a leaky echo state network, this study uses a network without a leak term because the fitting of the substorm occurrence probability was slightly degraded when the leak term was

introduced.

We represent the function $\nu$ as an exponential of the weighted sum of the state variables as

$$
\nu(t|\boldsymbol{\beta}) = \exp(\boldsymbol{\beta}^\mathsf{T} \boldsymbol{x}_k), \quad (t_k \le t < t_{k+1}).
\tag{7}
$$





Here, we have used the parameter vector $\boldsymbol{\beta}$ for the weights for determining the output of the ESN. The value of $\boldsymbol{\beta}$ is obtained with a Bayesian approach. We take the prior distribution of $\boldsymbol{\beta}$ as a Gaussian distribution as

$$p(\boldsymbol{\beta}) = \frac{1}{\sqrt{(2\pi\sigma^2)^m}} \exp\left[-\frac{\boldsymbol{\beta}^\mathsf{T}\boldsymbol{\beta}}{2\sigma^2}\right]. \tag{8}$$

where $m$ denotes the dimension of $\boldsymbol{x}_k$ and is 1000. We determine the standard deviation $\sigma$ based on the marginal likelihood (see Appendix) to avoid overfitting and underfitting. We set $\sigma = 0.6$ for analysing the *SML* substorm onsets in Section 3 and $\sigma = 1.4$ for analysing the *Wp* onsets in Section 4. We estimate $\boldsymbol{\beta}$ such that the following posterior probability density is maximised:

$$p(\boldsymbol{\beta}|\tau_{1:N}) = \frac{p(\tau_{1:N}|\boldsymbol{\beta})\,p(\boldsymbol{\beta})}{p(\tau_{1:N})}. \tag{9}$$

As $p(\tau_{1:N})$ does not depend on $\boldsymbol{\beta}$, we can obtain the optimal $\boldsymbol{\beta}$ by maximising the following objective function

$$
\begin{aligned}
J &= \log\left[p(\tau_{1:N}|\boldsymbol{\beta})\,p(\boldsymbol{\beta})\right] = \log L(\boldsymbol{\beta}) + \log p(\boldsymbol{\beta}) \\
&= \sum_{i=1}^{N} \log \nu(\tau_i|\boldsymbol{\beta}) - \sum_{k=0}^{K-1} \nu(t_k|\boldsymbol{\beta})\Delta t - \frac{\boldsymbol{\beta}^\mathsf{T}\boldsymbol{\beta}}{2\sigma^2} - \frac{m}{2}\log\left(2\pi\sigma^2\right)
\end{aligned}
\tag{10}
$$

using the Newton–Raphson method.

## 3 Analysis of auroral electrojet substorms

First, we analysed the SuperMAG substorm list by (Newell and Gjerloev, 2011a) derived from the SuperMAG data (Gjerloev, 2012). This SuperMAG substorm list identifies substorm onset times from the *SML* index, which indicates westward auroral electrojet intensity. The SuperMAG list thus enumerates the events in which a westward auroral electrojet was developed. We refer to such events as auroral electrojet substorms (AE substorms). We trained the ESN with data for the ten years from 2005 to 2014 so that the ESN output represents well the occurrence rate of the substorm onsets in the SuperMAG list. We used 5-minute values of the OMNI solar wind data as the input. As in NK22, we start the comparison after spin-up of the ESN for 72 steps in order that the ESN satisfactorily memorises the history of the input data. If more than half of the data were missing for 1 hour, we stopped the prediction and spun up the ESN again for the subsequent 72 steps. We then predicted the intensity function $\nu$ which represents instantaneous occurrence rate of substorm onsets.

Figure 2 shows an example of the prediction of the occurrence rate for the three days from 6 June to 8 June 2015. In the top panel, the red line indicates the occurrence rate $\nu$ (/hour) obtained with the ESN. The AE substorm onset times in the SuperMAG list are also plotted with blue triangles in this panel. For comparison, the *SMU* and *SML* indices, interplanetary magnetic field (IMF), solar wind speed, and solar wind density are shown in the second, third, fourth, and bottom panels, respectively. The IMF shown in the third panel is expressed in geocentric solar magnetic (GSM) coordinates, and the $x$, $y$, and $z$ components are indicated with the green, blue, and red lines, respectively. The predicted occurrence rate tended to increase when substorm onsets, indicated with blue triangles, were frequently observed. This means that the ESN predicted the occurrence pattern of the AE substorms well.



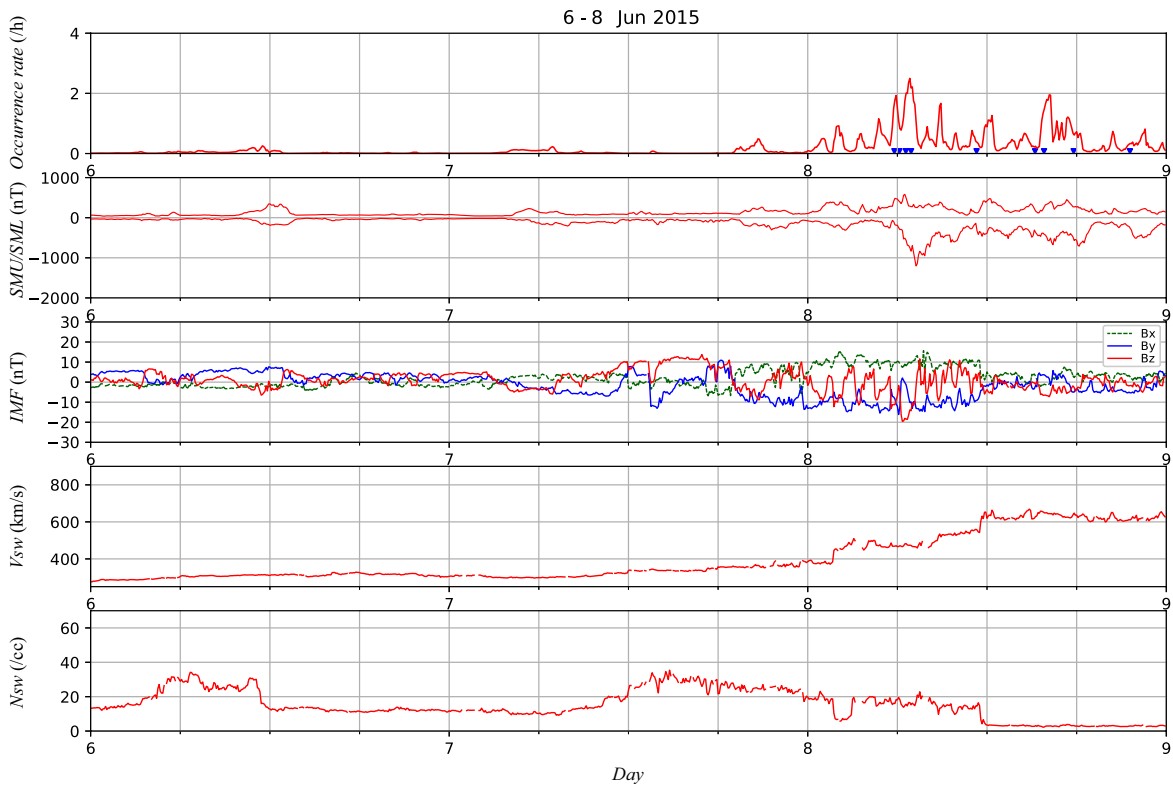

**Figure 2.** Predicted occurrence rate $\nu$ (/hour) for AE substorm onsets (top panel), the *SMU* and *SML* indices (second panel), IMF (third panel), solar wind speed (fourth panel), and solar wind density (bottom panel) for the three days from 6 June to 8 June 2015. The blue triangles in the first panel indicate the substorm onsets in the SuperMAG list. In the third panel, the $x$, $y$, and $z$ components in GSM coordinates are plotted with the green, blue, and red lines, respectively.

To assess the performance of the ESN-based prediction, we calculate the predicted probability of the substorm occurrence for each hour, obtained as

$$
P_k = 1 - \exp \left[ - \int_{t_k}^{t_k + 12\Delta t} \nu(t|\boldsymbol{\beta}) \, dt \right]
$$

$$
= 1 - \exp \left[ - \sum_{j=1}^{12} \nu(t_k + (j-1)\Delta t|\boldsymbol{\beta}) \, \Delta t \right]. \tag{11}
$$

Note that $12\Delta t$ corresponds to 1 hour because $\Delta t$ was taken to be 5 minutes. We classified the data into 10 classes by the predicted probability: $0 \le P_k < 0.1$, $0.1 \le P_k < 0.2$, ..., and $0.9 \le P_k$ and calculated the actual occurrence ratio for each class. Figure 3 shows the actual occurrence ratio with respect to the predicted probability for AE substorms in the four years from 2015 to 2018. For reference, the gray line indicates the case where the occurrence ratio is equal to the predicted probability.





The result with the red line shows that the predicted probability obtained with the ESN agrees well with the actual occurrence

ratio except that the prediction slightly overestimates the occurrence ratio for $0.4 \leq P_k < 0.8$.

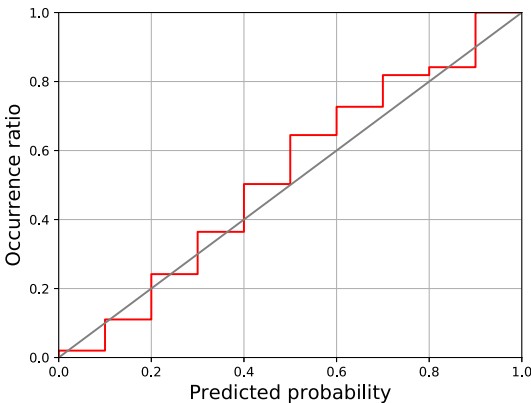

**Figure 3.** Actual occurrence ratio with respect to the predicted probability for AE substorms in the four years from 2015 to 2018 (red). The gray line shows the case where the predicted probability is equal to the actual occurrence ratio.

The performance of a probabilistic prediction is often evaluated with the Brier score, defined as

$$B = \frac{1}{K} \sum_{k=0}^{K-1} (r_k - P_k)^2, \tag{12}$$

where $r_k$ denotes the actual result. If the substorm occurred during the period $t_k \leq t < t_{k+1}$, $r_k = 1$, otherwise, $r_k = 0$. For example, with a non-informative prediction when the prediction $P_k$ is always 0.5, $(r_k - P_k)^2 = 0.25$ for the entire period

and the Brier score becomes $B = 0.25$. The Brier score gets smaller as the prediction improves. For the data from 2015 to 2018, the Brier score of the prediction with the ESN was 0.100. This Brier score should be compared with the case where we assume a stationary Poisson process where the occurrence rate $\nu$ is constant. If a stationary Poisson process is assumed and $\nu$ is optimised to fit the same data as used for training the ESN, the probability of substorm occurrence per hour is 15.6%. Using this value, the Brier score with the stationary Poisson process becomes 0.136, which is worse than the score with the ESN. The

better score with the ESN suggests that the ESN provides a meaningful prediction about the substorm occurrence.

## 4  Analysis of Pi2 substorms

In the following, we conduct a prediction of Pi2 substorms with the same approach as used in the previous section. This study identifies the Pi2 substorm onset using the *Wp* index (Nosé et al., 2009, 2012; World Data Center for Geomagnetism, Kyoto and Nosé, 2016). The *Wp* index corresponds to the amplitude of low-latitude Pi2 pulsations averaged over the nightside longitudes, which

is derived based on the wavelet analysis (Nosé et al., 1998). Nosé et al. (2012) proposed criteria for identifying Pi2 onsets from the *Wp* index. However, one of the criteria assumes quiescence of geomagnetic activity before the onset. To take into account Pi2 onsets during disturbed time, this study use a different method for identifying Pi2 onsets. Although the original *Wp* index





has a 1-minute resolution, our identification recipe uses moving averages with window sizes of five minutes to remove the effects of noisy oscillations. We detect a time of a peak of the moving averages, $T$, which satisfies the followings:

$$\overline{Wp}(T) > \overline{Wp}(T - 1\,\mathrm{min}), \tag{13}$$

$$\overline{Wp}(T) > \overline{Wp}(T - 2\,\mathrm{min}), \tag{14}$$

$$\overline{Wp}(T) \geq \overline{Wp}(T + 1\,\mathrm{min}), \tag{15}$$

$$\overline{Wp}(T) > \overline{Wp}(T + 2\,\mathrm{min}), \tag{16}$$

where $\overline{Wp}(T)$ denotes the moving average of the $Wp$ index at time $T$. We then find the time of the minimum $Wp$ within the 15-minute interval before the time of the peak and define it as $T - m\,\mathrm{min}\,(1 \leq m \leq 15)$. If $\overline{Wp}(T) - \overline{Wp}(T - m\,\mathrm{min}) > \theta\,\mathrm{nT}$, we identify this increase of $\overline{Wp}$ as a Pi2 event. To determine the onset time, we calculate the difference of $\overline{Wp}$ between five minutes apart:

$$\Delta\overline{Wp}(T - j\,\mathrm{min}) = \overline{Wp}(T - [j-2]\,\mathrm{min}) - \overline{Wp}(T - [j+2]\,\mathrm{min}) \tag{17}$$

for $j = 2, \ldots, m$. Finding $j = j^*$ which maximises $\Delta\overline{Wp}(T - j\,\mathrm{min})$, the time $T - j^*\,\mathrm{min}$ is regarded as the onset time. We hereinafter refer to the Pi2 events identified with the above criteria as Pi2 substorms, although they may contain pseudo-breakups and other phenomena which are not normally classified as substorms. In our event detection method, the threshold $\theta$ is a tunable parameter which determines the sensitivity of the detection. We set $\theta = 0.1\,\mathrm{nT}$ and considered two other cases: $\theta = 0.15\,\mathrm{nT}$ and $\theta = 0.2\,\mathrm{nT}$ for comparison, as described later.

We trained the ESN with data for the ten years from 2005 to 2014. Again, we used 5-minute values of the OMNI solar wind data as the input. We started the comparison after spin-up of the ESN for 72 steps. If more than half of the data were missing for 1 hour, we stopped the prediction and spun up the ESN again for the subsequent 72 steps. We then predicted the occurrence rate $\nu$ for substorm onsets. Figure 4 shows the prediction of the occurrence rate of Pi2 substorms for the three days from 6 June to 8 June 2015, which is the same event as in Figure 2. When identifying the substorm onsets, the threshold $\theta$ was set to be $0.1$ here. In the top panel, the red line indicates the occurrence rate (/hour) estimated with the ESN. The second panel shows the $Wp$ index, and the third panel shows the $SMU$ and $SML$ indices. The fourth, fifth, and bottom panels show the IMF in GSM coordinates, solar wind speed, and solar wind density, respectively. Similarly to Figure 2, the predicted occurrence rate tended to increase when Pi2 substorm onsets, indicated with blue triangles, were frequently observed.

We also calculate the predicted probability of substorm occurrence for each hour defined in Eq. (11). As in the previous section, we classified the data into 10 classes by the predicted probability: $0 \leq P_k < 0.1, \ldots, 0.9 \leq P_k$ and calculated the actual occurrence ratio for each class. Figure 5 shows the actual occurrence ratio with respect to the predicted probability for Pi2 substorms in the four years from 2015 to 2018. The predicted probability obtained with the ESN mostly agrees with the actual occurrence ratio although slightly underestimated. We also calculated the Brier score in Eq. (12). The Brier score values were 0.208, 0.180, and 0.150 with $\theta = 0.1$, $\theta = 0.15$, and $\theta = 0.2$, respectively. If a stationary Poisson process is assumed, the Brier score values were 0.236, 0.200, and 0.165 with $\theta = 0.1$, $\theta = 0.15$, and $\theta = 0.2$. The Brier score gets smaller as the threshold $\theta$ increases due to the number of events. If a larger threshold is taken, fewer events are identified as substorms. This



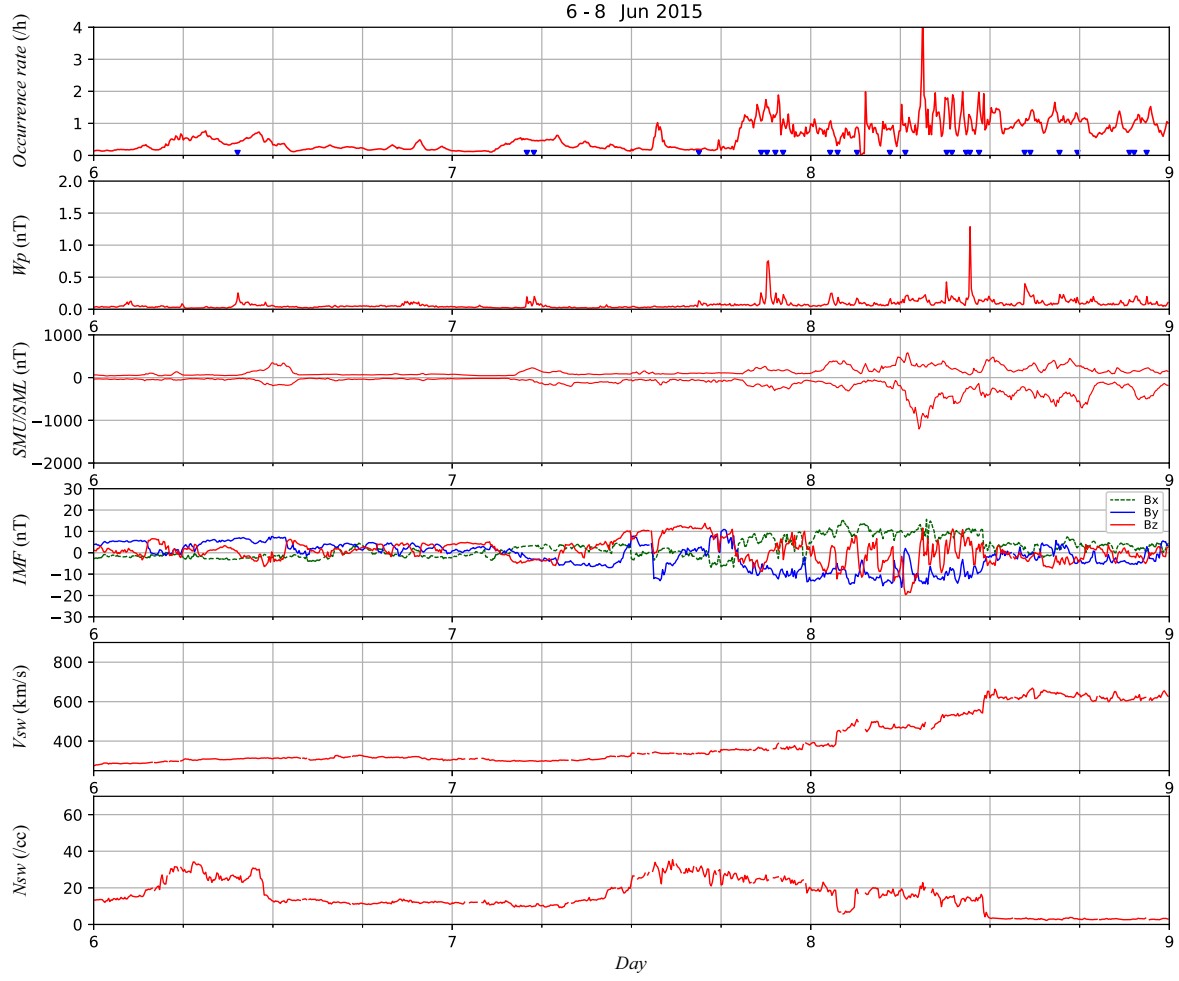

**Figure 4.** Predicted occurrence rate $\nu$ (/hour) for Pi2 substorm onsets identified from the $Wp$ index (top panel), the $Wp$ index (second panel), $SMU$ and $SML$ indices (third panel), IMF (fourth panel), solar wind speed (fifth panel), and solar wind density (bottom panel) for the three days from 6 June to 8 June 2015. The blue triangles in the top panel indicate the Pi2 substorm onsets identified from the $Wp$ index. The meaning of the colour in the fourth panel is the same as in the third panel of Figure 2.



situation tends to depress the predicted occurrence probability $P_k$. At the same time, as the number of events decreases, $r_k$ in Eq. (12) takes a value of 0 more frequently and $(r_k - P_k)^2$ accordingly decreases in more cases. Thus the Brier score depends on the number of events.

Comparing with the case of AE substorm onsets, the difference in the Brier score between the non-stationary and stationary Poisson process is likely to be smaller for Pi2 substorms. This presumably indicates that predicting Pi2 substorms is more difficult than AE substorms. Figure 6 is a histogram of the frequency of $P_k$ for Pi2 substorms identified with $\theta = 0.1$, Pi2 substorms identified with $\theta = 0.2$, and AE substorms. For AE substorms, $P_k$ was below $0.1$ in most cases. In contrast, it was relatively rare that $P_k$ was lower than $0.1$ for Pi2 substorms with $\theta = 0.1$. If $P_k$ approaches to 1, we can confidently predict the occurrence of a Pi2 substorm. If $P_k$ approaches to 0, we can confidently deny the occurrence of a Pi2 substorm. However, 195 $P_k$ for Pi2 substorms was between $0.2$ and $0.6$ for many cases, which means that it is usually difficult to confidently predict whether a Pi2 substorm occur or not. If the threshold is taken to be $\theta = 0.2$, the frequency of $P_k < 0.1$ increased but it was still much less than the case of AE substorms. Moreover, the frequency of $P_k \geq 0.7$ for Pi2 substorms with $\theta = 0.2$ is less than that for AE substorms. Overall, a Pi2 substorm onset seems to be less predictable than an AE substorm onset.

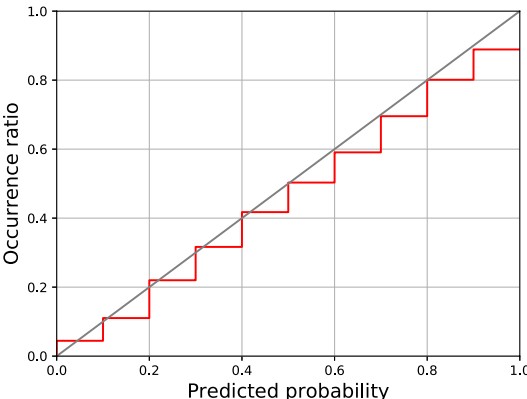

**Figure 5.** Actual occurrence ratio with respect to the predicted probability for Pi2 substorms in the four years from 2015 to 2018 with an occurrence threshold of $\theta = 0.15$ (red). The gray line shows the case where the predicted probability is equal to the actual occurrence ratio.

## 5 Response to synthetic solar wind

To determine what the ESN learned from the data, we conducted experiments analysing the response of the trained ESN to synthetic solar wind data. We used similar synthetic solar wind data to NK22 in which the solar wind parameters are fixed at constant values except that one of the parameters varying as a rectangular wave of various periods. Figures 7 and 8 demonstrate the experiments with the synthetic solar wind data over 18 days. The experiments in Figures 7 and 8 employ the ESN trained with the AE substorm data and with the Pi2 substorm data, respectively. In each figure, the top panel shows the predicted 205 probability of occurrence per hour, while the second, third, and fourth panels display the IMF, solar wind speed, and solar wind

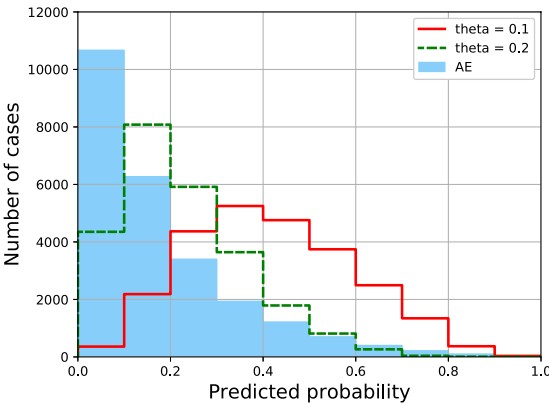

**Figure 6.** Histogram of frequency of $P_k$ for Pi2 substorms identified with $\theta = 0.1$ (red), Pi2 substorms identified with $\theta = 0.2$ (green), and AE substorms (blue) for the period from 2015 to 2018.

density, respectively. In both the experiments, the IMF $B_x$ and $B_y$ were set to 0 and the temperature was fixed at $5 \times 10^5 \, \text{K}$. In the first three days, IMF $B_z$ varied as a rectangular wave between $-5 \, \text{nT}$ and $1 \, \text{nT}$ with a period of 20 minutes for the first day, 2 hours for the second day, and 6 hours for the third day, while the solar wind speed was fixed at $400 \, \text{km/s}$ and the density was fixed at $2 \, / \text{cc}$. In the next three days (Day 4 to Day 6), IMF $B_z$ varied with the same pattern but the solar wind speed became

$600 \, \text{km/s}$. Next, IMF $B_z$ was fixed at $1 \, \text{nT}$ and the solar wind speed followed a similar rectangular pattern for three days (Day 7 to Day 9). After that, IMF $B_z$ became $-5 \, \text{nT}$ and the solar wind speed again followed the same rectangular pattern for three days (Day 10 to Day 12). The solar wind speed was then fixed at $600 \, \text{km/s}$, and the solar wind density was perturbed with a similar rectangular pattern under a fixed IMF $B_z$ of $1 \, \text{nT}$ (Day 13 to Day 15) and $-5 \, \text{nT}$ (Day 16 to Day 18).

     Both Figures 7 and 8 suggest that substorms tend to frequently occur when the IMF is southward and the solar wind speed is

high. Newell et al. (2016) discussed the importance of the solar wind speed based on the analysis of substorm onsets identified from the *SML* index. Figure 8 suggests that the Pi2 substorms are also strongly dependent on the solar wind speed. Indeed, the effect of the solar wind speed seems to be more essential for Pi2 substorms. From Day 4 to Day 6 when the solar wind speed was high, the occurrence rate of Pi2 substorms tended to be high even under northward IMF situations, while the frequency of AE substorms tended to be depressed by a northward IMF. This is interpreted as showing that Pi2 pulsations can occur under

geomagnetically quiet conditions without an intense auroral electrojet, which would be consistent with the result of Kwon et al. (2013). The existence of Pi2 pulsations under quiet conditions explains the poor predictability of Pi2 substorms suggested in Figure 6. Since AE substorms are rare during northward IMF, we can confidently deny the occurrence of a substorm when the IMF is northward. In contrast, since Pi2 substorms can often occur even when the IMF is northward, it would be rare that we can confidently deny the occurrence of a substorm.

In contrast with the response to the IMF and solar wind speed, the response to the solar wind density variation is spiky, especially for AE substorms (from Day 13 to Day 18, shaded with green in Figures 7 and 8). This may be interpreted as showing that a substorm is triggered by a sudden impulse (SI) of solar wind dynamic pressure, although it is possible that the SI





effect on the geomagnetic variation is misidentified as a substorm occurrence. It is also notable that a momentary enhancement of the occurrence rate is seen at the northward turning of the IMF on Day 6 for Pi2 substorms in Figure 8 (blue arrows in Figure 8). This might suggest that Pi2 substorms tend to be triggered by a northward turning of the IMF, as suggested by Lyons et al. (1997). However, when we tried the same experiment with the ESN trained with Pi2 substorms identified with a threshold of $\theta = 0.2$, the momentary enhancement of the predicted occurrence rate at the northward turning became unclear. Thus, the response to the northward turning of the IMF is not a distinct feature. As suggested by other studies (Morley and Freeman, 2007; Wild et al., 2009), the northward turning of the IMF is not likely to be essential to substorm occurrence even though the northward turning may give a favourable conditions for a substorm occurrence.

Excluding spikes of the substorm occurrence rate concurrent with jumps in the solar wind density, a higher solar wind density is likely to have increased the occurrence rate from Day 13 to Day 15 when the IMF was weak and northward. In contrast, from Day 16 to Day 18 when the IMF was southward, the occurrence rate did not show a clear change between before and after jumps of solar wind density except for spikes due to density jumps especially for AE substorms. These results suggest that the effect of solar wind density depends on the IMF. When the IMF is weak and northward, substorms tend to occur at a higher rate under higher solar wind density. Meanwhile, when the IMF is directed southward, the occurrence rate of substorms is less dependent on the solar wind density, though density jumps may affect the occurrence rate. This trend is similar to the compound effect of the solar-wind density and IMF on auroral electrojets as identified by NK22 and other studies (Ebihara et al., 2019; Kataoka et al., 2023). The solar-wind density effect on the occurrence rate may thus contribute to the dependence of the auroral electrojet intensity on the solar-wind density. However, according to NK22, a similar compound effect is also observed in the $AU$ index, which represents an eastward electrojet. Since eastward electrojets are not considered part of the substorm current system (e.g., Kamide and Kokubun, 1996), the occurrence rate cannot fully explain the compound effect on the auroral electrojets indicated by NK22.

## 6 Concluding remarks

This study proposed an approach for analysing event time series of substorm onsets with the occurrences influenced by solar wind conditions. We treat a substorm onset as a stochastic phenomenon and represent the stochastic occurrences of substorms with a nonstationary Poisson process. The occurrence rate of substorms is then described with an echo state network model with a time series of solar wind data as an input. The echo state network allows us to diagnose the relationship between the substorm occurrence rate and solar wind conditions. We applied this proposed approach to a sequence of AE substorms and a sequence of Pi2 substorms. The ESN successfully predicted the probability of occurrences of AE and Pi2 substorms.

We also conducted experiments with a synthetic solar wind data set and obtained the following results.

1. The occurrence rate is enhanced under a higher solar wind speed and southward IMF for both AE and Pi2 substorms. The solar wind speed effect is more important for Pi2 substorms than for AE substorms. While a high speed solar wind can bring Pi2 substorms even under a northward IMF, AE substorms are rarely observed under a northward IMF even if the solar wind speed is high.





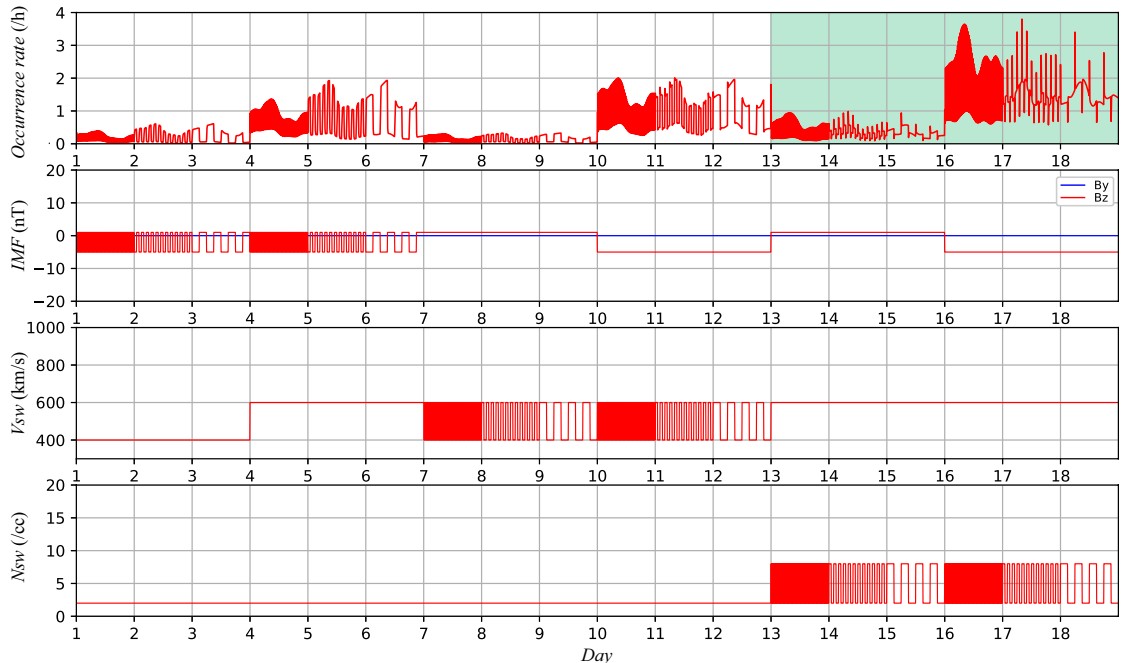

**Figure 7.** Occurrence rate $\nu$ (/hour) under synthetic data predicted with the ESN trained with the AE substorm data (top panel), the synthetic IMF (second panel), solar wind speed (third panel), and solar wind density (bottom panel). In the second panel, the $y$ and $z$ components in GSM coordinates are plotted with the blue, and red lines, respectively.

2. The occurrence rate is enhanced at an abrupt jump of the solar wind density. A northward turning of the IMF is also likely to momentarily enhance the occurrence rate of Pi2 substorms, although the effect of a northward turning seems to be minor.

3. A compound effect of the solar-wind density and IMF is also suggested. When the IMF is weak and northward, a higher solar wind density is likely to cause a higher occurrence rate of substorms. In contrast, when the IMF is southward, the solar wind density does not seem to make a clear effect on the occurrence rate of substorms except for spikes due to density jumps.

*Data availability.* The *AU* and *AL* indices are available from the web site of the WDC for Geomagnetism, Kyoto (http://wdc.kugi.kyoto-u.ac.jp/wdc/Sec3.html). The *SMU* and *SML* indices are available from the SuperMAG web site (https://supermag.jhuapl.edu). The SuperMAG substorm list are also be acquired through the SuperMAG web site. The Wp index is available via the web site of M. Nosé (https://www.isee.nagoya-u.ac.jp/~nose.masahito/s-cubed/). The OMNI solar wind data were acquired from the OMNIWeb of NASA/GSFC (https://omniweb.gsfc.nasa.gov/).



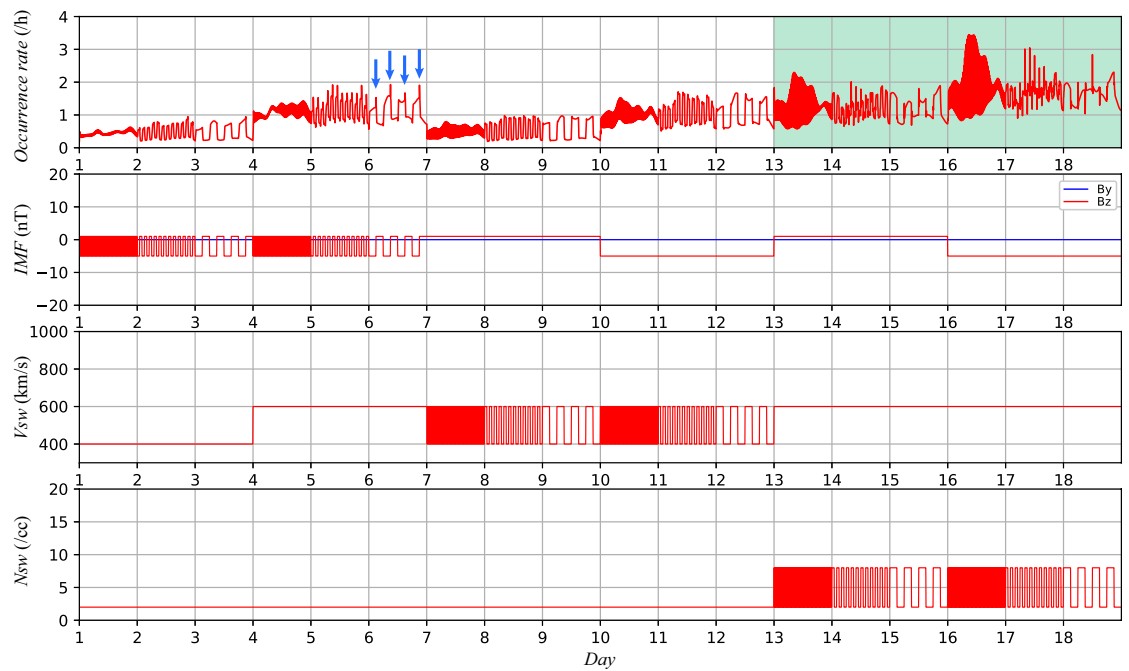

**Figure 8.** Occurrence rate $\nu$ (/hour) under synthetic data predicted with the ESN trained with the Pi2 substorm data (top panel), the synthetic IMF (second panel), solar wind speed (third panel), and solar wind density (bottom panel). In the second panel, the $y$ and $z$ components in GSM coordinates are plotted with the blue, and red lines, respectively.

## Appendix A: Parameter determination with marginal likelihood

Parameters of a Bayesian prior distribution is often determined based on the marginal likelihood (e.g., Morris, 1983; Casella, 1985). We determine the standard deviation of the prior distribution $\sigma$ by maximising the marginal likelihood which is defined as

$$p(\tau_{1:N}|\sigma) = \int p(\tau_{1:N}|\boldsymbol{\beta})\,p(\boldsymbol{\beta}|\sigma)\,d\boldsymbol{\beta} \tag{A1}$$

where we denote the prior distribution in Eq. (8) as the conditional distribution given $\sigma$, $p(\boldsymbol{\beta}|\sigma)$. Using Eq. (10), Eq. (A1) is reduced to

$$p(\tau_{1:N}|\sigma) = \int \exp[J]\,d\boldsymbol{\beta} \tag{A2}$$

Denoting the optimal value which maximising $J$ as $\hat{\boldsymbol{\beta}}$, it should satisfy $\nabla J(\hat{\boldsymbol{\beta}}) = 0$. The Taylor expansion of $J$ around $\hat{\boldsymbol{\beta}}$ is thus

$$J(\boldsymbol{\beta}) = J(\hat{\boldsymbol{\beta}}) + \frac{1}{2}(\boldsymbol{\beta} - \hat{\boldsymbol{\beta}})^{\mathsf{T}} \nabla^2 J(\hat{\boldsymbol{\beta}})(\boldsymbol{\beta} - \hat{\boldsymbol{\beta}}) + \cdots, \tag{A3}$$





where $\nabla^2 J(\hat{\boldsymbol{\beta}})$ is the Hessian matrix at $\boldsymbol{\beta} = \hat{\boldsymbol{\beta}}$. This Hessian matrix is negative definite because $J$ is maximised at $\boldsymbol{\beta} = \hat{\boldsymbol{\beta}}$. This second-order approximation of $J$ yields an approximation of Eq. (A1) as follows

$$
\begin{aligned}
&p(\tau_{1:N}|\sigma) \\
&\approx \int \exp\left[J(\hat{\boldsymbol{\beta}}) + \frac{1}{2}(\boldsymbol{\beta} - \hat{\boldsymbol{\beta}})^\mathsf{T} \nabla^2 J(\hat{\boldsymbol{\beta}})(\boldsymbol{\beta} - \hat{\boldsymbol{\beta}})\right] d\boldsymbol{\beta} \\
&= \frac{\sqrt{(2\pi)^m}\exp[J(\hat{\boldsymbol{\beta}})]}{\sqrt{|-\nabla^2 J(\hat{\boldsymbol{\beta}})|}},
\end{aligned}
\tag{A4}
$$

where $|-\nabla^2 J(\hat{\boldsymbol{\beta}})|$ is the determinant of the Hessian matrix of $-J$ at $\boldsymbol{\beta} = \hat{\boldsymbol{\beta}}$. This approximation is sometimes referred to as Laplace's approximation (e.g., Bishop, 2006). We chose the standard deviation $\sigma$ which maximises the logarithm of the approximate marginal likelihood:

$$
\log p(\tau_{1:N}|\sigma) = J(\hat{\boldsymbol{\beta}}) - \frac{1}{2}|-\nabla^2 J(\hat{\boldsymbol{\beta}})| + \frac{m}{2}\log 2\pi.
\tag{A5}
$$

*Author contributions.* SN conceived and conducted the analysis. RK contributed to the scientific interpretation. MN and JWG contributed to data processing and interpretation of the data.

*Competing interests.* The corresponding author declare that any authors have no competing interests.

*Financial support.* The work of SN was supported by JSPS KAKENHI (Grant Number 17H01704).

*Acknowledgements.* We acknowledge the substorm timing list identified by the Newell and Gjerloev techniqe (Newell and Gjerloev, 2011a), the *SMU* and *SML* indices (Newell and Gjerloev, 2011b), and the SuperMAG collaboration (Gjerloev, 2012).



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
