# Peer review of "Probabilistic modelling of substorm occurrences with an echo state network"

_Annales Geophysicae, 2023_

## Author Response (AR1)

**Response to the comments by Referee #1**

We appreciate the referee for the constructive comments and suggestions. We have revised the manuscript after carefully considering the comments raised by the referee. Some errors have also been fixed. In the following, the comments by the referee are quoted in Italic, and our reply is provided for each comment in Roman.

> *The question the current paper does not answer is: Does the new method give better results than existing methods, such as the method by Maimiti et al. (2019) or identification of substorm onsets from predicted AE indices? Such a comparison would be very useful, and in case larger statistics are considered to be too much work for the present paper, I'd like to see how the other methods perform in the example event at least.*

According to this suggestion, we tried to identify substorm onsets from the predicted SML index obtained with the echo state network (ESN) developed in our previous study (Nakano and Kataoka, 2022). As our previous study predicted the SML index with 5-minute resolution, we identified an onset using the following criteria:

1. $SML(t_0 + 5\mathrm{min}) - SML(t_0) < -45\,\mathrm{nT}$,

2. $SML(t_0 + 5\mathrm{min}) + SML(t_0 + 10\mathrm{min}) + SML(t_0 + 15\mathrm{min}) + SML(t_0 + 20\mathrm{min}) + SML(t_0 + 25\mathrm{min}) - SML(t_0) < -100\mathrm{nT}$,

which are similar to the criteria used by Newell and Gjerloev (2011). As a result, we identified only 246 onsets during the four years from 2015 to 2018, while we identified 4515 onsets from the 5-minute values of the actual SML index during the same period using the same criteria. As shown in Figure 1, many of negative SML spikes due to substorm onsets were not reproduced by the predicted SML index, and many of substorm onsets cannot be identified from the predicted SML index. We therefore believe that the proposed approach is more suitable for analysing substorm activity.

As the referee points out, Maimaiti et al. (2019) also predicted substorm onsets from solar wind data. However, their method addresses a slightly different task from our method. Maimaiti et al. predicted a substorm occurrence for next 60 minutes from the time history of solar wind data. They thus attempted to predict a substorm onset without using the solar wind data just before the onset. On the other hand, our purpose is to model the response to given solar wind inputs. We use the solar wind data with 5-minute resolution until the time interval of the onset to calculate the probability of the substorm occurrence. Another difference is in data selection. Maimaiti et al. selected the data so that the number of onset cases is equal to that of non-onset cases, which is favourable to attain a high F1 score. However, the number of onset cases is actually much less than that of non-onset cases. If a probabilistic model is trained by a data set in which onset cases and non-onset cases are equalised, it would provide a biased result when the occurrence rate is calculated. Our study uses the entire data from 2005 to 2014 except for spin-up periods due to data missing, which is appropriate to evaluate the occurrence rate of substorm onsets. It is therefore difficult to compare the two methods using the same metric.

In the revised version, we have improved an explanation on the difference from the approach of Maimaiti et al. (2019) (Line 42–45).

> *Lines 39-40: "a nonstationary Poisson process" Please provide a reference.*

A description on a nonstationary Poisson process is found in the textbook by Daley and Vere-Jones (2003). This textbook is cited in the revised version (Line 40).

> *Line 49: "DP2 type convection" Please provide a reference.*

The typical characteristics of the DP2 type convection is demonstrated by Nishida (1968). The description on the DP2 type convection is also found in the paper by Kamide and Kokubun (1996) which was already cited in the previous version. We have revised the sentence from Line 47 in the previous version (from Line 48 in the revised version) as follows:

"However, since the events determined from the auroral electrojet intensity may contain non-substorm events such as DP2 type convection enhancements (e.g., Nishida, 1968; Kamide and Kokubun, 1996), an increase of the auroral electrojet does not necessarily indicate a substorm onset."

> *Line 60: Please define "p".*

$p$ denotes the probability density. We have added the definition of $p$ (Line 62).

> *Line 81: "OMNI" Please provide a reference.*

In the revised version, we cite the online document by King and Papitashvili (2023) as a reference to the OMNI solar wind data (Line 83).

> *Line 274: "marginai" should be "marginal"*

We appreciate the correction. It has been corrected (Line 277).

> *Fig. 2 and 3: I suggest combining these figures to avoid repeating the same data and to make comparison of the two predictions easier.*
> *Fig. 7 and 8: I suggest combining these figures as well. What is the meaning of the shaded area? What are the arrows in Fig. 8? This information should be given in the caption.*

We thank the referee for the helpful suggestion. We think Figures 2 and 3 should not be combined because Figure 3 shows the analysis of Pi2 substorm onsets which are not yet explained in Section 3. However, we have combined Figures 7 and 8 according to this suggestion. The text has been edited accordingly. The caption in (new) Figure 7 has also been improved according to the referee's suggestion.

**References**

Daley, D. J. and Vere-Jones, D.: An introduction to the theory of point processes: Volume I: Elementary theory and method, 2nd ed., chap. 7, Springer, New York, 2003.

King, J. and Papitashvili, N.: One min and 5-min solar wind data sets at the Earth's bow shock nose, `https://omniweb.gsfc.nasa.gov/html/HROdocum.html`, last access: 21 August 2023.

Nishida, A.: Coherence of geomagnetic DP 2 fluctuations with interplanetary magnetic variations, J. Geophys. Res., 73, 5549–5559, 1968.

**Response to the comments by Referee #2**

We are grateful to the referee for the valuable comments. We have revised the manuscript after carefully considering the comments raised by the referee. Some errors have also been fixed. In the following, the comments by the referee are quoted in Italic, and our reply is provided for each comment in Roman.

> *L140-145: I would like to understand the importance of the difference of the Brier score. The Brier score of prediction with ESN is better than that with the stationary Poisson process. In a way, however, their values are also similar; the difference is 0.036. How different is the probability of substorm occurrence? In addition, could you also say that ESN provides a meaningful prediction about the substorm occurrence compared to other processes than the stationary Poisson process?*

The stationary Poisson process model assumes that the probability of substorm occurrence is constant. The probability is always rated at 15.6% with the stationary Poisson process model. In contrast, our ESN model assumes that the probability of substorm occurrence varies according to solar wind conditions. The frequency for each probability range is actually shown in Figure 6. It should be noted that, as indicated in Figure 6, the ESN model rated the probability for AE substorms to be less than 10% in nearly half of the cases. It was rare that the ESN rated the probability to be 50% or more. This suggests that it is difficult to confidently predict substorms. We thus interpret that the small difference in the Brier score between the stationary Poisson model and the ESN model indicates the difficulty in predicting substorms.

The referee also questions if we can compare the Brier scores between the ESN and other point process models. We understand there exists other point process models which consider the effects of past events on subsequent events. However, our ESN model does not consider the effects of past events because the main purpose of this paper is to model the relationship between solar wind conditions and substorm activity. Therefore, we think the comparison with the stationary Poisson process model, which ignores the time history of past events, is sufficient for evaluating the effectiveness of our approach which represents the solar wind effects using the ESN.

To clarify our purpose of the comparison of the Brier score, we will revise the last sentence of Section 3 as follows:
"The better score with the ESN confirms that the information on the solar wind, which is used as the input for the ESN, effectively improves the prediction about the substorm occurrence."

> *Figure 4: The occurrence rate appears to enhance where Wp index has no enhancement (i.e., 7 June 2015, around 14 UT). Is this because other effect such as SI effect on the geomagnetic variation is included?*

In our ESN model, the occurrence rate is predicted from time history of the solar wind data. We interpret that the enhancement of the predicted occurrence rate around 14 UT on 7 June 2015 was caused by the variation of the IMF.

The enhancement of the predicted occurrence rate does not guarantee an enhancement of the Wp index. It just means that the solar wind condition is favourable for Pi2 occurrence. It is

thus possible that the predicted occurrence rate increases without any Pi2 onset.

*L240-244: I could not understand this argument. Do you mean that the occurrence rate does not change with the solar wind density?*

We intended to say that the dependence of the occurrence rate on solar wind density varies according to the IMF conditions.

When the IMF is weak and northward, the substorm occurrence rate is higher when solar wind density is higher. Meanwhile, when the IMF is southward, the dependence of the occurrence rate on the solar wind density is not clear except that the solar wind density jumps can affect the occurrence rate. We have modified the expression in the revised version (Line 243–245).

*Figure 5, 6: Please specify that the predicted probability is 'Pk' in these figures so that readers can easily follow what predicted probability (Pk) means in both figure and text.*

We appreciate the suggestion. We now specify that the predicted probability is $P_k$ in Figure 3, 5, and 6 in the revised version.